# Efficacy of Fasting in Type 1 and Type 2 Diabetes Mellitus: A Narrative Review

**DOI:** 10.3390/nu15163525

**Published:** 2023-08-10

**Authors:** Daniel Herz, Sandra Haupt, Rebecca Tanja Zimmer, Nadine Bianca Wachsmuth, Janis Schierbauer, Paul Zimmermann, Thomas Voit, Ulrike Thurm, Kayvan Khoramipour, Sian Rilstone, Othmar Moser

**Affiliations:** 1Division of Exercise Physiology and Metabolism, BaySpo—Bayreuth Center of Sport Science, University of Bayreuth, 95447 Bayreuth, Germany; daniel.herz@uni-bayreuth.de (D.H.); sandra.haupt@uni-bayreuth.de (S.H.); rebecca.zimmer@uni-bayreuth.de (R.T.Z.); nadine.wachsmuth@uni-bayreuth.de (N.B.W.); janis.schierbauer@uni-bayreuth.de (J.S.); paul.zimmermann@uni-bayreuth.de (P.Z.); thomas.voit@uni-bayreuth.de (T.V.); thurm@idaa.de (U.T.); sian.rilstone@uni-bayreuth.de (S.R.); 2Department of Cardiology, Klinikum Bamberg, 96049 Bamberg, Germany; 3Interdisciplinary Center of Sportsmedicine Bamberg, Klinikum Bamberg, 96049 Bamberg, Germany; 4Department of Physiology and Pharmacology, Afzalipour School of Medicine, Kerman University of Medical Sciences, Blvd. 22 Bahman, Kerman 7616914115, Iran; k.khoramipour@gmail.com; 5Department of Metabolism, Digestion and Reproduction, Imperial College London, London SW7 2BX, UK; 6Interdisciplinary Metabolic Medicine Trials Unit, Department of Internal Medicine, Division of Endocrinology and Diabetology, Medical University of Graz, 8036 Graz, Austria

**Keywords:** diabetes mellitus, fasting interventions, intermittent fasting, time-restricted fasting, periodic fasting, Ramadan fasting

## Abstract

Over the last decade, studies suggested that dietary behavior modification, including fasting, can improve metabolic and cardiovascular markers as well as body composition. Given the increasing prevalence of people with type 1 (T1DM) and type 2 diabetes mellitus (T2DM) and the increasing obesity (also in combination with diabetes), nutritional therapies are gaining importance, besides pharmaceutical interventions. Fasting has demonstrated beneficial effects for both healthy individuals and those with metabolic diseases, leading to increased research interest in its impact on glycemia and associated short- and long-term complications. Therefore, this review aimed to investigate whether fasting can be used safely and effectively in addition to medications to support the therapy in T1DM and T2DM. A literature search on fasting and its interaction with diabetes was conducted via PubMed in September 2022. Fasting has the potential to minimize the risk of hypoglycemia in T1DM, lower glycaemic variability, and improve fat metabolism in T1DM and T2DM. It also increases insulin sensitivity, reduces endogenous glucose production in diabetes, lowers body weight, and improves body composition. To conclude, fasting is efficient for therapy management for both people with T1DM and T2DM and can be safely performed, when necessary, with the support of health care professionals.

## 1. Introduction

The prevalence of diabetes is increasing, with the global incidence expected to rise from 436 million in 2019 and expected to expand to 578 million in 2030 and 700 million in 2045 [1]. This is a result of increasing incidence of both type 1 (T1DM) [2] and type 2 diabetes mellitus (T2DM) [1].

The combination of insulin resistance, insufficient insulin secretion, and excessive glucagon secretion can cause T2DM and may lead to micro- and macrovascular complications [3]. In response to the increasing number of people with T2DM and the rising prevalence of obesity, the American Diabetes Association (ADA) and the European Association for the Study of Diabetes (EASD) have assembled a panel to update consensus statements on the treatment of T2DM [4]. Lifestyle changes via improved physical activity and nutrition have become a prioritized treatment for people with T2DM and should be promoted as a primary intervention [4].

Interestingly, individuals with T1DM face many problems seen in those with T2DM, with elevated body weight resulting in deteriorated insulin sensitivity, and as a consequence, are considered to have “double diabetes” [5]. Considering the increased rates of overweight and obesity in individuals with T1DM [6], there is an urgent need to explore the potential of nutritional interventions for T1DM.

As part of the multifactorial approach to managing diabetes, more emphasis is on weight loss, which is essential for the treatment of T2DM through individualized therapeutic interventions. This includes dietary recommendations that promote healthy food choices and eating behaviors and consider personal preferences to establish sustainable healthy eating habits [4]. Furthermore, an energy deficit via fasting is one of many options that can be explored for long-term weight loss in T2DM [4]. Recent research has suggested that fasting can enhance insulin sensitivity and glucose tolerance in individuals who are overweight and at high risk of developing T2DM [7].

Published research assessing the effect of fasting in T1DM has demonstrated positive outcomes, including a reduced need for exogenous insulin [8], stabilization of glycemia, a reduction in body weight and body mass index (BMI), and lower total carbohydrate intake [9]. However, there are limited data on the safety of fasting, particularly on the risk of hypoglycemia and diabetic ketoacidosis (DKA). Therefore, this review aims to provide an overview of the safety and efficacy of various fasting interventions in people with T1DM and T2DM and to provide a basis for fasting in diabetes.

## 2. Materials and Methods

A narrative, non-systematic review of the literature was conducted in September 2022. We searched via PubMed using the following combined text and Medical Subject Heading terms: “fasting interventions”, “intermittent fasting”, “time-restricted feeding”, “periodic fasting”, “Ramadan fasting”, and “diabetes mellitus.” The words were combined with the Boolean operators AND or OR. Studies potentially relevant to the research question were considered for this review. The reference lists of the studies found were also evaluated to identify additional studies. The PubMed database search initially yielded 27,485 citations, of which 97 references were considered for the review.

## 3. State-of-the-Art of Intermittent and Periodic Fasting

Intermittent fasting (IF), including alternate day fasting (ADF), time-restricted feeding (TRF), and periodic fasting (PF), can be considered an energy intake reduction method [10,11,12]. These fasting procedures reduce daily calorie intake by 75% to 100% during fasting, which is followed by a feasting period [10,13]. Several IF protocols have gained popularity [14]; however, it is unlikely that all IF diets will result in homogenous physiological adaptation since the pattern of fasting and eating are performed differently [15]. Intermittent fasting generally involves a fasting window of 12 to 72 h [10]. The most common IF can be divided into groups as follows (some of these regimens are classified as TRF, which is usually performed daily) [16]: the 16:8 method consists of 16 h of fasting followed by an 8 h feeding window [17]. The feeding window can be shortened to 4 h in a more rigorous approach, the so-called 20:4 method [17]. Another protocol is ADF, consisting of a 24 h fasting period alternating with a 24 h eating period, repeated two or three times a week [18,19]. Also widely used is twice-per-week fasting (TWF) (e.g., the so-called 5:2 diet), which involves a very low-calorie diet two days a week (consecutive or non-consecutive), following ad libitum feeding for the other five days [16]. Another form of IF is fasting from a religious perspective, for example, Ramadan fasting (RF). Healthy Muslim youths and adults have to fast for 28–30 days [20] from sunrise to sunset [21]. The fasting period involves abstinence from food intake, hydration, sexual activity, and smoking [22].

Intermittent fasting decreases body weight and fat mass and improves blood pressure and insulin sensitivity [23,24]. The results of other studies suggest that TRF in young men reduces calorie intake (without the need for calorie counting) and a significant reduction in fat mass without affecting lean mass [25]. In addition, ADF is a highly effective IF pattern that has been shown to reduce fat mass in healthy individuals, while fat-free mass is unchanged and has a positive impact on cardiovascular markers [26]. Research has shown that weight loss can also be achieved through the TWF pattern, where individuals eat and drink usually for five days and restrict calorie intake for two non-consecutive days per week [27]. These trends have been replicated in a study in which healthy young women’s body weight and BMI were reduced during RF [28].

Periodic fasting involves an extended and significant calorie-restricted diet, often allowing only water for the duration of the fast, typically from 48 h to one week [11]. Intermittent fasting usually extends over 12 to 48 h and takes place regularly, i.e., every 1 to 7 days. Periodic fasting, on the other hand, extends over 2 to 7 days and takes place no more than once a month [29]. There are two primary methods of PF: (a) water-only fasting, which involves consuming only water during the fasting period, and (b) a “fasting-mimicking diet”, which is a low-calorie, plant-based diet that is low in protein and sugar and high in unsaturated fat [29].

Periodic fasting provides numerous benefits, including weight loss, improved insulin sensitivity, glycemia, and reduced circulating lipids [30,31]. In addition, PF can also activate autophagy and promote cell renewal [32].

## 4. Physiological Pathways of Fasting

The metabolic processes in the human feeding–fasting and sleep–wake cycles are influenced by a complex network of circadian pacemakers [33]. Daily sleep and activity rhythms depend on a complex interaction between endogenous cell-autonomous molecular oscillators and exogenous factors such as daily exposure to light/darkness and the feeding/fasting cycle [31,34]. Acquiring and storing food during periods of availability and utilizing these resources during fasting periods without compromising health and fitness are essential to the 24 h rhythms [31,35].

Circadian clocks play an essential role in glucose and lipid metabolism by causing fluctuations in circulating hormone levels in response to various stimuli [36]. For example, the production of hormones such as melatonin or cortisol depends on responses to light and darkness, while other hormones are regulated by the feeding and fasting cycle [36]. While melatonin initiates the resting phase, cortisol is responsible for the body’s activity phase [36]. Other hormones like fibroblast growth factor 21 (FGF21), adiponectin, and leptin influence glucose and lipid metabolism by promoting either catabolism with fatty acid oxidation and glycolysis or anabolism with lipogenesis and glycogenesis [36]. The circulating levels of glucagon and insulin in the body are influenced by various endocrine factors that exhibit diurnal fluctuations [33]. The timing of these fluctuations is crucial for optimal physiological processes [37]. The molecular mechanisms responsible for altered meal patterns’ effects on metabolic health through fasting are partly related to the synchronization between fasting time and circadian rhythms [38]. Fasting can be an important modulator of circadian rhythm [39]. Fasting can impact the circadian rhythm by altering meal timing, affecting the timing and amplitude of various physiological processes [31].

Fasting triggers a switch in metabolism mediated by significant changes in several metabolic signaling pathways [40]. These metabolic adaptations are characterized by increased circulating ketones bodies [41] while circulating fatty acids, amino acids, glucose, and insulin are maintained at low concentrations [42]. In addition, fasting with reduced carbohydrate intake leads to the breakdown of liver glycogen, mobilization of fatty acids from adipose tissue, and stimulation of β-oxidation in the liver with increased production of ketones bodies (β-hydroxybutyrate) and acetyl–coenzyme A (AcCoA) [16]. The production of ketone bodies, such as β-hydroxybutyrate from a fatty acid breakdown, can act as endogenous histone deacetylase inhibitors and contribute to epigenetic control of gene expression, DNA repair, and genome stability [40].

During fasting, the body’s energy needs are met by breaking down stored fat and glycogen [42]. This process results in a decrease in insulin levels, which in turn leads to a decrease in protein breakdown or proteolysis [42,43,44,45]. This reduction in proteolysis helps to preserve muscle mass during fasting periods [46]. However, it is worth noting that the correlation between fasting and reduced proteolysis may depend on various factors, such as the length of the fast, the individual’s nutritional status, and the amount and intensity of physical activity performed during the fasting period [43,44], Figure 1.

Adaptive cellular responses from periods of fasting include an increase in adenosine monophosphate (AMP) and adenosine diphosphate (ADP) and a decrease in cellular adenosine triphosphate (ATP), leading to the activation of the AMP-activated protein kinase (AMPK) [42]. The activation of AMPK inhibits various anabolic metabolic pathways, stimulates catabolic reactions, and stimulates autophagy, characterized by the removal of damaged proteins and a “rejuvenation” of organelles, and results in an improvement of mitochondrial function [47]. When there is a decrease in circulating amino acids and glucose, it inhibits mammalian targets of Rapamycin (mTOR) activity, resulting in reduced protein synthesis to conserve energy [48] and increased mitochondrial biogenesis [40]. This, in turn, may lead to a decrease in oxidative stress and an increase in autophagy [16]. Regular fasting cycles also have systemic anti-inflammatory effects and can increase the number of progenitor stem cells. The downregulation of insulin-like growth factor 1 (IGF-1) and the reduction of circulating amino acids suppress mTOR activity, which is involved in regulating inflammatory processes in the body, and reduced mTOR activity has been linked to decreased inflammation [48].

Additional benefits of fasting include increased brain-derived neurotrophic factors (BDNF) and lowered systolic and diastolic blood pressure by activating the parasympathetic system [17]. Ketogenesis also promotes synaptic plasticity and neurogenesis by increasing the expression of BDNF [48]. Brain-derived neurotrophic factors cause the release of acetylcholine by the vagus nerve, which reduces the frequency of resting heart rate [17]. Studies on the development of atherosclerosis have confirmed other effects of fasting on the cardiovascular system. Fasting has been shown to reduce the concentration of inflammatory markers such as interleukin 6, homocysteine, and C-reactive protein [49] (Figure 2). In conclusion, the mechanisms discussed here are the most thoroughly researched and widely recognized effects of fasting, bearing in mind that there may be other intricate pathways that have not yet been explored in equal detail.

## 5. Impact of Fasting as an Adjuvant Therapy in T1DM and T2DM

### 5.1. Fasting and Its Effects on T1DM

Pathophysiological processes of T1DM have a direct impact on the individual’s metabolism [50], the gut microbiome [51], cognitive performance [52], and related cardiovascular pathways [53]. The therapy of T1DM focuses on preventing complications by adjusting glycemia with exogenous insulin, using technological support, dietary changes, and exercise [54]. Additionally, since the loss of functional β-cell mass is one of the critical aspects, various therapeutic approaches have been developed in recent decades to prevent or slow down this process [55]. To preserve β-cell mass, and replace endogenous insulin, exogenous insulin is used as the primary treatment for all people with T1DM [56]. Whilst conventional therapy existing of fixed-dose regimens was standard practice in the past and can still be used in certain circumstances, it is generally not the preferred option [57]. Multiple rapid-acting insulin analogs are frequently administered before meals, along with one or more daily separate injections of long-acting insulins [58].

Given the increasing prevalence of overweight and obesity in children, adolescents, and adults with T1DM, adjuvant therapeutic options may be considered [6]. For longstanding T1DM, insulin resistance to exogenous insulin may worsen, which may be associated to some degree with body weight [59]. Between the Diabetes Control and Complications Trial (DCCT) and the Epidemiology of Diabetes Interventions and Complications (EDIC) trial, it was observed that excessive weight gain in DCCT was linked to insulin resistance, dyslipidemia, elevated blood pressure, and atherosclerosis during the EDIC trial [60]. It is hypothesized that in T1DM, continuous exposure to exogenous insulin is inversely related to lipolysis rates. Therefore, safe, feasible strategies for managing body weight and insulin resistance need to be defined [60]. In this context, fasting has been demonstrated to be adjuvant support for individuals with T1DM, decreasing exogenous insulin [8].

Based on previous studies (Table 1), it can be inferred that fasting in T1DM can attain positive results when insulin adjustments are accounted for. The frequency and time spent in hypoglycemia must remain unaffected [61] or ideally reduced [62]. Furthermore, various forms of fasting can improve glycemia [59], decrease body weight and increase the quality of life [9]. Additionally, the studies show that the hemoglobin A1c levels (HbA1c) value can decrease after fasting interventions [8] or during RF [63]. In addition to the positive results of metabolic changes, fasting interventions can adversely affect the cardiometabolic system and body composition [9].

However, fasting is not without risk in people with T1DM. Uncontrolled fasting can lead to kidney damage, including acute kidney injury due to dehydration exacerbated by concomitant use of nephrotoxic drugs [62] and hypoglycemia if endogenous mechanisms for counter-regulation cannot keep up with energy expenditure [67]. During Ramadan, there was a significant increase in the frequency of severe hypoglycemia compared to other months, with individuals with T1DM experiencing 0.14 episodes per month versus 0.03 episodes per month in other months [64]. For children with T1DM, it is strongly recommended that they do not fast due to the high risk of acute complications such as hypoglycemia and potentially DKA [68]. However, there is very little evidence to suggest that DKA is more prevalent during Ramadan [65]. Hypoglycemia is a common concern during RF, but it can be prevented by adopting diabetes self-management education and support principles. Using technology, such as continuous glucose monitoring during Ramadan, could help people with T1DM identify hypoglycemic and hyperglycemic episodes complicated by medication adjustment or omission during fasting [68]. Despite the positive effects of fasting and its potential to improve cardiovascular outcomes, there are limited data on the impact of fasting on safety, i.e., hypoglycemia and DKA. Furthermore, changes in the required insulin dose after a prolonged fasting period in people with T1DM have not been adequately studied [9].

### 5.2. Fasting and Its Effects on the Prevention of T2DM

Pre-diabetes is the preliminary stage of T2DM, which in most cases, leads to the development of T2DM [69]. It is often associated with obesity, dyslipidemia, high triglycerides, and/or low high-density lipoprotein, and hypertension [70]. Pre-diabetes may be preventable through targeted lifestyle and weight loss interventions and can therefore help to avoid the development of T2DM [7]. Fasting can be applied as an intervention to counteract body weight increase and thus the progression of possible pathophysiology that can lead to T2DM. A review by Cienfuegos et al. summarized that TRF resulted in modest weight loss (1 to 4% from baseline) and energy restriction when food intake was restricted [7]. In addition, it also lowered fasting insulin levels, increased insulin sensitivity in people with pre-diabetes and obesity [7], improved glucose tolerance, and reduced serum glucose excursions. Possible mechanisms underlying these benefits included increased autophagic flux, a slight increase in ketone bodies, a reduction in oxidative stress, and stimulation of β-cell responsiveness [71].

Given the association of T2DM with obesity, elevated free fatty acids (FFA) levels lead to insulin resistance by inhibiting the insulin signaling pathway [72]. The insulin signaling pathway is an intracellular pathway that is triggered by the binding of insulin to its receptor on the surface of target cells and ultimately regulates various cellular processes such as glucose uptake, glycogen synthesis, protein synthesis, and lipid metabolism [72].

Some new strategies, which focus on the timing of food intake and the duration of fasting, are becoming more prominent for weight loss and the accompanying improvement in metabolic health in people with T2DM. Carter et al. [73] showed that continuous or intermittent caloric restriction resulted in a significant reduction in HbA1c, with no difference between treatment groups (−0.5% [0.2%]) in the continuous energy restriction group versus (−0.3% [0.1%]) in the intermittent energy restriction group. In this study, 97 out of 137 participants with T2DM completed an assigned intervention phase. Participants were divided into two groups: an intermittent energy restriction diet (500–600 kcal/d), which was carried out on two non-consecutive days per week (on the other five days, participants ate as usual), or a continuous energy restriction diet (1200–1500 kcal/d), which was carried out seven days per week for 12 months. In general, a reduction in HbA1c levels in T2DM through fasting has been confirmed in several studies [3,74,75,76]. Obermayer et al. reported that IF could lead to a remarkable reduction in HbA1c (−7.3 ± 12.0 mmol/mol) over 12 weeks, in contrast to the control group (0.1 ± 6.1 mmol/mol). No cases of severe hypoglycemia were observed [76]. As summarized by Joaquim et al. [77], further metabolic benefits have been explained in T2DM: Activation of AMPK increases fat oxidation and glucose uptake by the GLUT4 transporter, which improves insulin sensitivity. Activation of the GLUT4 transporter leads to its translocation from intracellular vesicles to the plasma membrane, resulting in increased glucose uptake by the cell.

Li et al. [78] demonstrated that people with T2DM who underwent a one-week prolonged fasting program using Buchinger’s method showed a mean 3.5 kg reduction in body mass, decreased abdominal circumference and blood pressure, and improved quality of life. The program involved consuming only 300 kcal/day of liquids and gradually reintroducing solid food.

### 5.3. Pharmacotherapy and Fasting in T2DM

Treating people with T2DM is challenging, with a significant focus on pharmacological therapy. Individualized therapy is recommended, and concomitant diseases need to be considered. If HbA1c targets are not achieved even with the second line of treatment, a third line of treatment can be added [77]. Combining medications for T2DM with fasting may pose a potential risk, especially regarding hypoglycemia.

Many antidiabetic medications have their clinical effects through different mechanisms [77]. For metformin, Corley et al. [79] assessed whether the risk of hypoglycemia in people with T2DM was greater if they followed a very low-calorie diet for 2 consecutive days compared to a very low-calorie diet for 2 non-consecutive days while eating ad libitum for the remaining 5 days, with ongoing metformin medication. Participants with a BMI of 30–45 kg/m^2^ and a HbA1c of 50–86 mmol (6.7–10%) treated with metformin and/or blood glucose-lowering medication were enrolled. Over 12 weeks, the average hypoglycemia rate was 1.4 events over the 12 weeks. During the fasting days, the dosage of sulfonylureas and isophane insulin (NPH) decreased by 50%. The dosage of insulin lispro (Humalog), insulin aspart (Novorapid), insulin glulisine (Apidra), and regular insulin (Humulin R) was reduced by 70% on days of fasting. Mixed insulins were decreased by 25% the night before and 50% on the day of a fast. The dosage of insulin glargine (Lantus) was decreased by 50% in the morning or the evening before a fasting day. The dosages of metformin or other medications that do not cause hypoglycemia were kept the same. Although the medication was reduced, fasting increased hypoglycemia, with no difference between consecutive and non-consecutive day fasting. Despite this, both group participants experienced improved weight, fasting glucose, HbA1c levels, and overall quality of life. There appears to be a low risk of taking metformin in combination with fasting in people with T2DM [80].

In addition to metformin, the action of incretin can treat T2DM and obesity simultaneously. Incretin-based therapies utilize the action of a glucose-dependent insulinotropic polypeptide (GIP) and glucagon-like peptide 1 (GLP-1). Glucagon-like peptide 1 receptor agonists (GLP-1Ras), among other indications, are recommended as an add-on therapy for those who do not reach the required HbA1c threshold after three months of metformin therapy [81]. Stimulation of GLP-1Ras in the satiety centers of the hypothalamus is thought to be responsible for the appetite reduction observed with GLP-1Ras [82]. Because of the minimal risk of hypoglycemia with GLP-1Ras and the presumed effect on the appetite center, GLP-1Ras, in combination with TRF, are thought to have a minimal risk of hypoglycemia [83].

A sodium–glucose co-transporter-2 (SGLT-2) is responsible for most glucose reabsorption in the proximal renal tubules [84]. As its inhibition leads to a reduction in blood glucose levels, there is a potential benefit for treating T2DM [84]. Because of the beneficial effects of inhibiting SGLT-2, a new strategy is to develop dual sodium–glucose co-transporter-1 (SGLT-1)/SGLT-2 inhibitors. Sodium–glucose co-transporter-1 is the primary transporter of glucose uptake in the gut and is also expressed in the proximal renal tubule [85]. These dual drugs reduce glucose absorption in the gastrointestinal tract by inhibiting SGLT-1 and renal glucose reabsorption by inhibiting both transporters [84,85]. The risk of hypoglycemia is low because the effect of SGLT-2 inhibitors is not dependent on insulin [86]. Furthermore, with SGLT2 inhibitors, there is an increased production of ketone bodies, especially in fasting phases or when blood glucose levels are low. These enhanced ketone bodies can be attributed to increased fatty acid oxidation and ketogenesis via the liver [87]. There is evidence that SGLT2 inhibitors may increase the consumption of ketone bodies in the heart, leading to potential cardioprotective effects. The heart can use ketone bodies as an efficient source of energy, and during periods of fasting or low glucose availability, the heart may rely more heavily on ketones for energy. In addition, some studies have shown that SGLT2 inhibitors can improve heart function and reduce the risk of cardiovascular events in people with T2DM [88,89,90].

Complementary to pharmaceutical treatment, dietary changes, including caloric restriction and fasting, can be used to prevent and manage T2DM [91]. Careful monitoring of glycemia during fasting is the most important aspect for patients taking pharmacological agents that can cause hypoglycemia [92]. The management of T2DM during other types of fasting has already been discussed, emphasizing the selection and adjustment of oral hypoglycemic agents concerning HbA1c levels [93]. Common antidiabetic drugs such as metformin, GLP-1Ras, or SGLT-2 can be used with fasting. Based on the existing literature, fasting does not appear to pose significant safety risks for people with T2DM if medication is carefully monitored and adjusted as needed [83].

## 6. Discussion

Although the evidence on fasting as adjuvant therapy in diabetes is limited, it suggests notable benefits. Time-restricted feeding has been shown to improve glucose and fat metabolism and increase insulin sensitivity by directly regulating insulin and glucagon [71]. Increased insulin sensitivity is enhanced by the expression of adiponectin, which sensitizes adipose tissue to insulin and leptin and mimics the effect of insulin monotherapy [94].

In addition to improving glycemia during RF [63], studies indicate that PF reduces BMI and improves the quality of life in T1DM [9]. Fasting can significantly reduce the rates of hypoglycemia in T1DM [66]. However, care should be taken during RF, especially in children with T1DM [65,68]. Additionally, studies have shown that IF can decrease HbA1c values in T1DM [8,63]. However, the fasting phase in T1DM should always be performed under medical supervision. In people without diabetes, the processes described above are regulated by a delicate balance of circulating insulin levels and counter-regulatory hormones that help maintain glucose concentrations within the physiological range. However, in diabetes, glucose homeostasis is disrupted by the underlying pathophysiology and often by pharmacological agents designed to enhance or supplement insulin secretion [93]. In people with T1DM, glucagon secretion may not increase sufficiently in response to hypoglycemia. Furthermore, in some individuals with T1DM, adrenaline secretion is also impaired due to a combination of autonomic neuropathy and defects associated with recurrent hypoglycemia [95]. In T1DM with absolute insulin deficiency, prolonged fasting without adequate insulin can lead to excessive glycogen depletion and increased gluconeogenesis and ketogenesis, resulting in hyperglycemia and DKA [93]. People with T1DM who fast during Ramadan are at increased risk of developing DKA or hypoglycemia [64,65,68], mainly if their diabetes is poorly controlled before Ramadan [93]. Clinical trials including participants with T2DM consistently showed a reduction in fasting insulin levels, an effect primarily associated with weight loss. Several physiological responses to TRF are similar to those of regular aerobic exercise. These include improved insulin sensitivity and cellular stress, as well as reduced resting blood pressure and heart rate [31]. The benefits of IF may be mediated by highly conserved nutrient recognition pathways that control the circadian rhythm of appetite and energy expenditure and stress response pathways that upregulate cellular defenses, such as antioxidant enzymes [77].

Fasting has been shown to have a positive impact on T2DM. Time-restricted feeding protocols are more convenient and safer than other fasting protocols, as patients eat on a daily basis, and fasting periods are not long enough to significantly increase the risk of hypoglycemia [92]. Besides hypoglycemia, the main adverse effects are psychological and physiological disorders such as anxiety and muscle wasting. Furthermore, IF may have an additional impact on insulin dosing in people with T2DM [96]. Multidisciplinary treatment can mitigate the most adverse effects, including the psychological and physiological distress of the diet.

In terms of “double diabetes”, there is evidence to suggest that fasting can improve both insulin sensitivity and glucose metabolism, which may have beneficial effects on individuals with “double diabetes”.

One potential cellular mechanism linking insulin resistance and glucose metabolism is inhibiting insulin signaling pathways [72]. Insulin resistance can occur due to impaired signaling through these pathways, which can lead to reduced glucose uptake and impaired metabolism [72]. Fasting has been shown to activate various signaling pathways, including AMPK and sirtuins, leading to insulin sensitivity and glucose metabolism [42].

Overall, both “double diabetes” and fasting are associated with changes in insulin resistance and glucose metabolism at the cellular level and thus may theoretically establish a relationship between fasting and double diabetes at the cellular level. Further research is needed to fully understand the underlying mechanisms and develop targeted interventions to improve glucose control in individuals with “double diabetes”.

Given the increasing prevalence of T1DM, T2DM, and obesity, developing therapy strategies is crucial. This also requires long-term studies that can demonstrate the potential for the prevention of adverse health outcomes. The type of protocol used is also critical, as TRF or moderate energy restriction has shown better results than stricter protocols, which are usually associated with increased oxidative stress and fasting glycemia. Furthermore, given the large number of Muslims who fast during Ramadan, it is of considerable importance to individually consult people with diabetes to reduce hypoglycemia during RF. In addition, further studies are needed that explain the relationship between the most common medications used to treat T1DM and T2DM and fasting.

In order to be able to assess the efficiency of fasting over a longer period of time, for people with diabetes, it is increasingly recommended to use diabetes technology. For example, continuous glucose monitoring (CGM) systems provide the opportunity to be used as a real-time “motivational device” to support patients during interventions like fasting. Integrating CGM systems into fasting allows people with T1DM or T2DM to closely monitor their glucose levels. Using real-time data, patients can make informed decisions about their fasting programs and adjust their approach according to how their glucose levels respond to fasting [76]. Complementary to CGM systems, closed-loop systems combine CGM with insulin pumps to create an automated insulin delivery system. These systems use CGM data to adjust insulin delivery in real-time, reducing the risk of hypoglycemia during fasting and keeping blood glucose levels more stable. By integrating closed-loop systems into fasting interventions, especially for those with T1DM, patients can fast in a safer and more controlled way, especially during PF [97].

## 7. Conclusions

Our review provided an overview of the current research on whether fasting can be considered an adjuvant therapy for T1DM and T2DM. We showed that individuals with T1DM and T2DM may safely undertake a fasting period with a limited risk of severe glycaemic disturbances. However, future studies of greater duration and using different fasting methods are needed to capture the full range of potential benefits of fasting in people with T1DM and T2DM.

## Figures and Tables

**Figure 1 nutrients-15-03525-f001:**
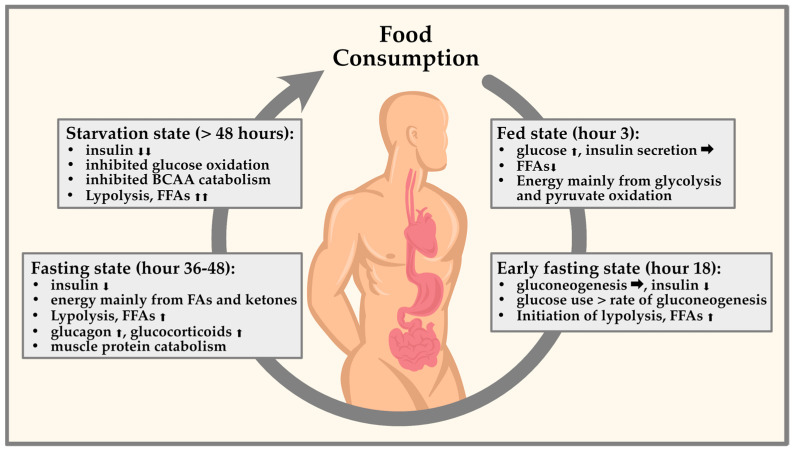
Representation of circulating glucagon and insulin in fasting periods. Taking the circadian rhythm into account, various endocrine factors are determined by diurnal fluctuations. These fluctuations are important for physiological processes to occur at the optimal time of day. Fasting already leads to changes in glucose and lipid metabolism in the first 48 h by promoting either catabolism with fatty acid oxidation and glycolysis or anabolism with lipogenesis and glycogenesis.

**Figure 2 nutrients-15-03525-f002:**
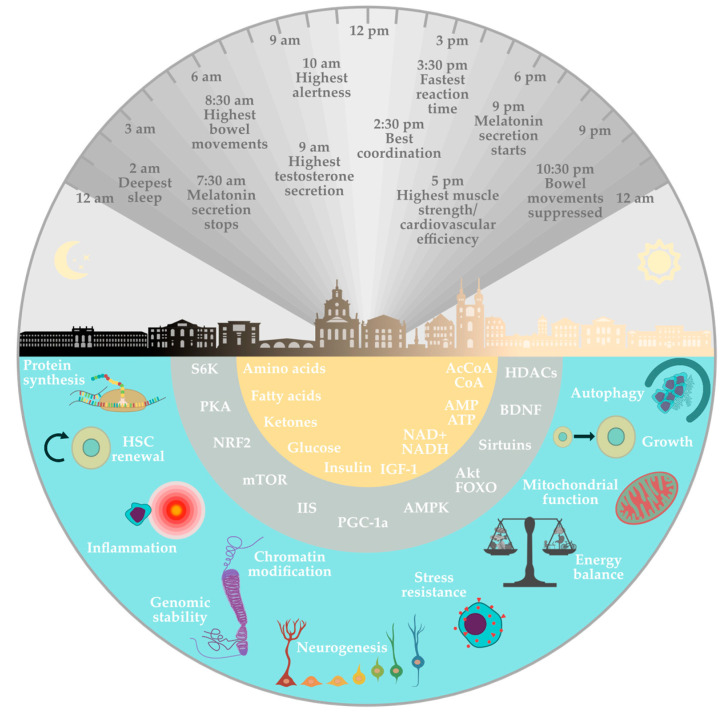
Physiological mechanisms involved in the health effects of fasting. The lower semicircles refer to the relative effect of fasting on the different metabolic pathways involved. The upper semicircle represents the relationship between the circadian rhythm and some individual physiological effects at a given time of day within 24 h.

**Table 1 nutrients-15-03525-t001:** The selected studies show an overview of associations between fasting and its effects on T1DM. Abbreviations as in the legend to Table 1, and: RF, Ramadan-Fasting; TCH, total cholesterol; LDL, low-density lipoprotein; HbA1c, hemoglobin A1c; IGF-1, insulin-like growth factor 1; OGTT, oral glucose tolerance test; BMI, body mass index; T1DM, type 2 diabetes mellitus; Ø, the event did not occur.

Author and Year	Sample Size	Fasting Regime	Results
Al-Ozairi et al. (2019) [62]	*n* = 43	RF	↓ hypoglycemia
El-Hawary et al. (2016) [63]	*n* = 53 (children)	RF	↓ fructosamine
			↑ TCH and LDL
			↓ HbA1c
Reiter et al. (2007) [8]	*n* = 43	prolonged fasting (>25 h)	↓ insulin dosage
			↑ HbA1c
Hassanein et al. (2020) [61]	*n* (<18 years) = 370 *n* = 279 (fasted)	RF	holistic reported hypoglycemia (60.7%)
	*n* (>18 years) = 1113 *n* = 761 (fasted)		holistic reported hyperglycemia (44.8%)
Salti et al. (2004) [64]	*n* = 11,173	RF	holistic reported hypoglycemia (0.14 episodes/Ramadan vs. 0.03 episodes/other months)
Mohamed et al. (2019) [65]	*n* = 50	RF	Fasting on 20 ± 9.9 days in Ramadan (7.8% broke the fast due to mild hypoglycemia)
Berger et al. (2021) [9]	*n* = 20 (with T1DM)	7-day fasting	ketoacidosis Ø
	*n* = 10 (without disease)		blood glucose was stable (4.9 (±1.5) and 7.5 (±2.3) mmol/L)
			↑ quality of life
			↓ BMI
			↑ diastolic blood pressure 69.75 (±11.41) to 75.74 (±8.42) mmHg
Moser et al. (2021) [66]	*n* = 20 (with T1DM)	(a) overnight fasting—12 h fasting (b) prolonged fasting—36 h fasting	occurred hypoglycemia during the night period 12 h.: 0.07 ± 0.06 rate per hour vs. 36 h: 0.05 ± 0.03 rate per hour
			fat oxidation 12 h: 90 ± 40 g/day vs. 36 h: 130 ± 35 g/day
			carbohydrates oxidation 12 h: 305 ± 98 g/day vs. 36 h: 215 ± 63 g/day
			Body weight 12 h: 76.7 ± 13.5 kg vs. 36 h: 75.4 ± 13.4 kg
			BMI 12 h: 24.6 ± 2.8 kg/m^2^ vs. 36 h: 24.2 ± 2.8 kg/m^2^

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
