# Peer review of "Efficacy of Fasting in Type 1 and Type 2 Diabetes Mellitus: A Narrative Review"

_nutrients, 2023, doi:10.3390/nu15163525_

Round 1

Reviewer 1 Report

The main issue I have with this review is that I cannot differentiate clearly the authors' opinions like in a perspective/editorial from the conclusions derived from their literature review. There is description of how publications were searched and then there is a table 1 with selected studies reviewed, but not clear findings of this literature search; e.g. how many found; how many rejected, main findings, etc.

As it stands now this is simply a narrative of opinions on the subjects with references to the literature.

Author Response

Response to Reviewer 1 Comments

We would like to thank the reviewer for evaluating our manuscript and for the constructive feedback. We highly appreciate the reviewer comments and tried to incorporate them into the revised version of the manuscript.

General remark:

All modifications/corrections are marked (using modification mode/”redlining” in Word) and additionally highlighted (yellow) in the text.

Point 1: “The main issue I have with this review is that I cannot differentiate clearly the authors’ opinions like in a perspective/editorial from the conclusions derived from their literature review. There is description of how publications were searched and then there is a table 1 with selected studies reviewed, but not clear findings of this literature search; e.g. how many found; how many rejected, main findings, etc.

As it stands now this is simply a narrative of opinions on the subjects with references to the literature.”

Response 1: We thank the reviewer for this methological comment. Since this paper is a narrative review as also give already in the title of the paper, we have refrained from exactly presenting the results of the literature review, as otherwise it would be a systematic review, e.g. according to the PRISMA scheme – which was not defined as our aim. The most important findings of the literature review were considered and reviewed from the authors' point of view. In the method section, the total number of references and the number of references included were changed as follows (see lines 79-80): “The PubMed database search initially yielded 27485 citations, of which 100 references were considered for the review.”, resulting from the input of keywords including Boolean operators. The content of the current literature that was essential for the research question of this paper were taken into account in the preparation and our own opinion was exlcuded per se. For further information, we used the follwing recommendations for our review: https://familymedicine.med.wayne.edu/mph/project/green_2006_narrative_literature_reviews.pdf

Reviewer 2 Report

I read with great interest the manuscript "Efficacy of Fasting in Type 1 and Type 2 Diabetes Mellitus: A Narrative Review" by Herz et al.

The paper is fine. It is logically divided into sections and subsections. English is fine, only minor spell check. I really like figures.

Comments:

1. Introduction, line 56-57, "Individualized therapies that target bodyweight and glycaemia are crucial for managing T2DM": this is limited. In fact, many studies have proven that a correct managment is a multifactorial approach. We are diabetologist, not glycaemologist. In fact, by losing weight we act on several pathways that lead to an amelioration of patient's condition due to improvement of all other risk factors, incuding gycaemia.

2. Pharmacotherapy and fasting: "Glucagon-like peptide 1 receptor agonists (GLP-1Ras) are recommended as add-on therapy for those who do not reach the required HbA1c threshold after three months of metformin therapy", I would add "in addition to other indications".

3.As SGLT2i improve ketone body consumption in the heart, is there any evidence about an interplay on this event between fasting and SGLT2i.

4. I would add a future perspective little paragraph.

fine

Author Response

Response to Reviewer 2 Comments

We would like to thank the reviewer for evaluating our manuscript and for the constructive feedback. We highly appreciate the reviewer comments and tried to incorporate them into the revised version of the manuscript.

General remark:

All modifications/corrections are marked (using modification mode/”redlining” in Word) and additionally highlighted (yellow) in the text so that all changes made are understandable.

Point 1: English is fine, only minor spell check.

Response 1: We thank the reviewer for the comment. Our paper was again assessed for grammar and language – we hope this if fine now.

Point 2: Introduction, line 56-57, "Individualized therapies that target bodyweight and glycaemia are crucial for managing T2DM: this is limited. In fact, many studies have proven that a correct management is a multifactorial approach. We are diabetologist, not glycaemologist. In fact, by losing weight we act on several pathways that lead to an amelioration of patient's condition due to improvement of all other risk factors, including glycaemia.”

Response 2: We thank the reviewer for this important comment. We have carefully reviewed the note and corrected it accordingly, it has been amended as follows (see lines 56-58): “As part of multifactorial approach to manage diabetes, more emphasis is on weight loss, which is essential for the treatment of T2DM through individualised therapeutic interventions. Furthermore, we want to clarify that our idea of using the word “individualized” does include especially the meaning “multifactorial.”

Point 3: Pharmacotherapy and fasting: "Glucagon-like peptide 1 receptor agonists (GLP-1Ras) are recommended as add-on therapy for those who do not reach the required HbA1c threshold after three months of metformin therapy", I would add "in addition to other indications".

Response 3: We thank the reviewer for this very important comment. We have amended this sentence as recommended by the reviewer (see lines 325-328): “Glucagon-like peptide 1 receptor agonists (GLP-1Ras), among other indications, are recommended as add-on therapy for those who do not reach the required HbA1c threshold after three months of metformin therapy.”

Point 4: As SGLT2i improve ketone body consumption in the heart, is there any evidence about an interplay on this event between fasting and SGLT2i.

Response 4: We do much appreciate this important comment. With regard to the reviewers’ comment, we have included some essential aspects on the interplay between fasting, SGLT2 inhibitors and the resulting ketone body consumption in the heart (see lines 344-352): “Furthermore, with SGLT2 inhibitors there is an increased production of ketone bodies, especially in fasting phases or when blood glucose levels are low. This enhanced ketone bodies can be attributed to increased fatty acid oxidation and ketogenesis via the liver [89]. There is evidence that SGLT2 inhibitors may increase the consumption of ketone bodies in the heart, leading to potential cardioprotective effects. The heart can use ketone bodies as an efficient source of energy, and during periods of fasting or low glucose availability, the heart may rely more heavily on ketones for energy. In addition, some studies have shown that SGLT2 inhibitors can improve heart function and reduce the risk of cardiovascular events in people with T2DM [90–92].”

Point 5: I would add a future perspective little paragraph.

Response 5: We thank the reviewer for this recommendation. Now we have included some perspectives on fasting interventions and the huge relevance of the use of technologies such as CGM or closed-loop systems in people with diabetes (see lines 425-438): “In order to be able to assess the efficiency of fasting over a longer period of time, for people with diabetes also it is increasingly recommended to use diabetes technology. For example, continuous glucose monitoring (CGM) systems provide the opportunity to be used as a real time “motivational device” to support patients during interventions like fasting. Integrating CGM systems into fasting allows people with T1DM or T2DM to closely monitor their glucose levels. Using real-time data, patients can make informed decisions about their fasting programmes and adjust their approach according to how their glucose levels respond to fasting [99]. Complementary to CGM systems, closed-loop systems, combine CGM with insulin pumps to create an automated insulin delivery system. These systems use CGM data to adjust insulin delivery in real time, reducing the risk of hypoglycaemia during fasting and keeping blood glucose levels more stable. By integrating closed-loop systems into fasting interventions especially for those with T1DM, patients can fast in a safer and more controlled way, especially during PF [100].”

Round 2

Reviewer 2 Report

The authors appropriately answered to all the issues I raised.